# Design and Fabrication of Dual-Scale Broadband Antireflective Structures on Metal Surfaces by Using Nanosecond and Femtosecond Lasers

**DOI:** 10.3390/mi11010020

**Published:** 2019-12-24

**Authors:** Rui Lou, Guodong Zhang, Guangying Li, Xuelong Li, Qing Liu, Guanghua Cheng

**Affiliations:** 1State Key Laboratory of Transient Optics and Photonics, Xi’an Institute of Optics and Precision Mechanics of CAS, Xi’an 710119, China; lourui0423@163.com (R.L.); li_guang_ying@163.com (G.L.); 2University of Chinese Academy of Sciences, Beijing 100049, China; 3School of Future Technology, University of Chinese Academy of Sciences, Beijing 100049, China; 4Electronic Information College, Center of Optical Imagery Analysis and Learning (OPTIMAL), Northwestern Polytechnical University, Xi’an, Shaanxi 710072, China; zhangguodong@opt.cn (G.Z.); li@nwpu.edu.cn (X.L.); 5Zhejiang Wanli University, Ningbo 315100, China; liuqing@opt.ac.cn

**Keywords:** antireflective surfaces, nanosecond laser, femtosecond laser, micro-nano hybrid structures, broadband spectrum

## Abstract

Antireflective surfaces, with their great potential applications, have attracted tremendous attention and have been the subject of extensive research in recent years. However, due to the significant optical impedance mismatch between a metal surface and free space, it is still a challenging issue to realize ultralow reflectance on a metal surface. To address this issue, we propose a two-step strategy for constructing antireflective structures on a Ti-6Al-4V (TC4) surface using nanosecond and femtosecond pulsed lasers in combination. By controlling the parameters of the nanosecond laser, microgrooves are first scratched on the TC4 surface to reduce the interface reflection. Then, the femtosecond laser is focused onto the sample surface with orthogonal scanning to induce deep air holes and nanoscale structures, which effectively enhances the broadband absorption. The antireflection mechanism of the dual-scale structures is discussed regarding morphological characterization and hemispherical reflectance measurements. Finally, the modified sample surface covered with micro-nano hybrid structures is characterized by an average reflectance of 3.1% over the wavelengths ranging from 250 nm to 2250 nm.

## 1. Introduction

Surface micro- and nanostructuring of metal, with potential applications in antireflection [1,2,3,4,5,6,7,8], transdermal enhancement [9], and wettability alternation [10,11], has gained great interest from researchers in recent years. Various related techniques have been proposed to produce regulated structures on metal surfaces. Hierarchical micro- and nanostructures inspired by nature were recently designed and demonstrated to have multiple functions in metal surface modification, thus drawing significant attention. The hierarchical structures can effectively address the optical impedance mismatch between metal and free space, achieving ultralow reflection over the broadband spectrum on a metal surface. For instance, ZnO substrates covered by biomimetic hierarchical structures exhibit low reflection over a wide spectral range [12]. Due to their excellent performance in surface antireflection, such hierarchical structures have been applied in various fields, such as artificial blackbodies, photodetectors, infrared imaging, stray light shields, space-borne optics, etc. [13,14,15,16].

Several techniques for fabricating micro- or nanostructures have been proposed, such as chemical etching [17], coating [18], reactive ion etching [19], and pulsed laser processing [20,21]. The chemical etching method is one of the most widely used techniques; it uses chemical etchants to etch the surface of the materials to form a rough surface, but there is poor control over the shape of the structures. The coating method uses a light-absorbent material attached to the substrate to restrain reflection, which has the disadvantage of the coating material easily detaching from the substrate. The reactive ion etching method is performed by using molecular gas plasma in a vacuum system to fabricate structures; this is a relatively complex process and generates mechanical stress and damages the material.

Comparing with the above techniques, laser processing on the metal surface has many merits, including contactless treatment, scratch resistance, no accelerated fading, digital control, and environmental friendliness [22]. Many advances have been made in antireflective surface laser texturing and many pioneering works have been completed. Direct ultrafast laser nanostructuring was used to prepare biomimetic omnidirectional antireflective glass [23]. Ripples were formed on a stainless-steel surface using a femtosecond laser to enhance absorption [24]. Periodic black silicon antireflection surface structures were fabricated using four-beam laser interference lithography and studied [25]. In general, nanosecond pulsed lasers have proved to be a reliable tool for achieving ultralow reflectance due to their high processing efficiency and low cost [26,27]. Boinocivh et al. used a nanosecond laser to induce regular microgrooves by covering titanium alloy surfaces with porous nanostructures, and the modified surfaces were superior in terms of both corrosion and wear resistance [28]. Periodic microarrays covered with bushy nanoparticles on a nickel surface were fabricated using a 100-ns pulsed laser, which significantly enhanced the optical absorption [29]. However, due to the great thermal effects and collateral damage induced by thermal melting, nanosecond laser structuring is suboptimal in constructing nanoscale features on a sample interface. In contrast, ultrafast laser processing, with its advantages of short laser duration and modification process (speeding up before significant thermal diffusion), is usually considered as nonthermal processing, thereby enabling the convenient creation of nanoscale structures on a sample surface, and it has gradually become a popular choice in surface modification in recent years. Zheng et al. used a hybrid method consisting of ultrafast-laser-assisted texturing and chemical fluorination treatment to fabricate a black titanium alloy by efficiently enhancing the surface broadband antireflection [30]. Nevertheless, the relatively low processing efficiency of ultrafast laser systems makes the search for a creative strategy to ameliorate this shortcoming in previous work urgent [31,32,33].

In this work, to combine the advantages of ultrafast laser fabrication and nanosecond laser fabrication, we present a two-step fabrication process using nanosecond and femtosecond lasers on Ti-6Al-4V (TC4) surfaces, which improves the work efficiency of the processing. Microstructures and nanostructures were fabricated by the nanosecond and the femtosecond lasers, respectively, to enhance the metal surface broadband absorption. The morphology and formation mechanism of the hierarchical micro- and nanostructures were carefully studied.

## 2. Experimental Setup

### 2.1. Materials

A Ti-6Al-4V (TC4) titanium alloy sample with a size of 25 mm × 25 mm × 10 mm was used as the sample in our experiments. The top surface of the TC4 was mechanically polished (MPD-1, MOWEI, China) and cleaned ultrasonically (PS-20, Jeken, China) with ethanol before laser treatment.

### 2.2. Preparation of the Micro-Nano Hybrid Structures

A two-step processing technique was designed to induce hierarchical micro- and nanostructures on the TC4 surface. An illustration of the process is shown in Figure 1. First, the micro-scale structures were fabricated on the polished TC4 surface using the nanosecond laser (BX-2-G, Edgewave, Germany) processing system as demonstrated in Figure 1a. The laser source had a wavelength of 532 nm and a pulse duration of 14 ns. The pulse repetition rate was fixed at 100 kHz and the pulse energy varied from 50 μJ to 200 μJ in the experiment. The laser, passing through a scanning galvanometer (GO5, JCZ Technology, China), was focused on the TC4 surface with a focus diameter of about 10 μm. The calculated fluence value ranged from 127 J/cm^2^ to 509 J/cm^2^, much higher than the ablation threshold of 1.7 J/cm^2^ for a nanosecond laser [34]. The scanning galvanometer enabled the laser focus to move at a high speed. Arrays of microgrooves were scratched conveniently on the sample surface. The influence of laser parameters, including the scanning interval, scanning velocity, and pulse energy, on surface modification was carefully studied.

The femtosecond laser processing system was used to induce nanoscale structures and air holes in the microgrooves with an orthogonal scratching direction, further improving the antireflection properties of the sample surface. The femtosecond laser source (Pharos, Light Conversion, Lithuania) had a wavelength of 1030 nm and a pulse duration of 800 fs. The pulse repetition rate was fixed at 100 kHz and the pulse energy was fixed at 5 μJ. The laser, passing through a 10× objective, was directly focused with a diameter of 3.4 μm on the sample surface in an atmospheric environment. The calculated fluence value was about 110 J/cm^2^, much higher than the ablation threshold of 1.7 J/cm^2^ for a femtosecond laser [35]. A three-dimensional mobile platform (ANT, Aerotech, USA) was utilized to control the laser scanning direction and velocity.

### 2.3. Measurement and Characterization

The sample surfaces covered with hierarchical micro- and nanostructures were characterized using field emission scanning electron microscopy (SEM, JSM-7500F, Japan). The hemispherical reflectance of a sample over the UV, visible, and NIR regions (250–2250 nm) was measured using a spectrophotometer (Lambda 1050, PerkinElmer, USA) incorporated with an integrating sphere that was 150 mm in diameter. 

## 3. Results and Discussion

Surface optical absorption is strongly dependent on the material dielectric functions, surrounding medium, and surface morphologies [36]. For a certain sample, it is convenient to regulate the absorption property through surface structuring. The hierarchical micro- and nanostructures were demonstrated to be able to effectively enhance the surface absorption of a sample over the broadband spectrum. The regime can be explained by using a Fresnel-equation-based mathematical model that describes the behavior of light propagating between media with different refractive indices. When light reaches the interface between different optical media, all or part of the light will be reflected. The reflectance R is then calculated as [37,38]:(1)$$R={\left[\frac{\sqrt{n_{2}^{2}-{\left(n_{1}\sin\theta \right)}^{2}}-n_{1}\cos\theta }{\sqrt{n_{2}^{2}-{\left(n_{1}\sin\theta \right)}^{2}}+n_{1}\cos\theta }\right]}^{2}$$
for the incident case from the air, where *n*_1_, approximately equal to 1, is the refractive index of air; *n*_2_ is the refractive index of the solid substrate; and *θ* is the incident angle of light waves. When the light is at normal incidence, i.e., θ = 0°, the reflectance *R* is given by:(2)$$R={\left(\frac{n_{2}-1}{n_{2}+1}\right)}^{2}$$

Figure 2 shows a schematic diagram of light waves propagating on sample surfaces with different morphologies. In Figure 2a, when the sample surface is polished to be flat enough, part of the light is absorbed by the substrate material, and the other part is reflected. The absorption is mainly influenced by the intrinsic characteristics of the substrate material. When the sample surface is covered with micro- or nanostructures, the light wave propagation will be complicated, as illustrated in Figure 2b,c. If the feature size of the surface structure is much larger than the wavelength of the incident light, the microstructures will effectively capture the incident light, and the light is reflected back to the surface. As a result, multiple internal reflections will occur to improve the coupling of light with the substrate material, which will eventually lead to light being absorbed into the microstructures inside. The microholes acting as light traps can reduce the surface reflection through this internal multireflection process. The micro-nanostructured surface not only generates both optical coupling and an optical path increase as a result of the microstructure characteristics, but also has the “moth-eye effect” resulting from the nanostructure characteristics. Hierarchical micro-nano structures are more effective for antireflection than microstructures alone.

### 3.1. Effect of Nanosecond Laser Scanning Interval on Surface Reflectance

A focused nanosecond laser with a high intensity in the GW/cm^2^ range can induce laser ablation and restructure the metal surface. Considering the relatively large pulse duration (>10^−9^ s), the laser processing is accompanied by apparent thermal effects, and the thermal diffusion length $2\sqrt{{D\tau }_{\mathrm{dura}}}$ can be on the micron scale. The morphologies of the microstructures fabricated on the TC4 surface using a nanosecond laser are shown in Figure 3a–e. Figure 3a–d show samples with intervals of scanning lines ranging from 30 μm to 90 μm, while Figure 3e corresponds to a local magnification of Figure 3b. Generally, a transition from the normal vaporization mechanism to a violent ejection of a mixture of vapor and liquid droplets from the irradiated target (usually called phase explosion or, more appropriate, explosive vaporization) occurs at a threshold laser irradiance of the solid material. If fracture or sufficient evaporation occurs, the surface may disintegrate and ablate in the subsequent process [39,40]. It can be seen that a large amount of laser-melted ejections formed in the laser-irradiated region. When the scanning interval was decreased, the microgrooves gradually disappeared and transformed into randomly distributed concavo–convex microstructures. This is attributed to the overlap between adjacent laser-irradiated lines. Since the width of the laser-modified line was larger than 50 μm, subsequent line-scanning influenced and reshaped the morphology of the previous scanning line. The concavo-convex microstructures crowded together and form microholes on the sample surface, as shown in Figure 3b. The multi-reflection and scattering processes on the sample surface were enhanced in this case. When the interval of welding lines was further decreased to 30 μm, the ablation process slightly smoothed the concavo-convex microstructures; therefore, the number and depth of microholes showed an apparent decrease. 

The reflection spectra of the samples over the wavelength from 250 nm to 2250 nm are shown in Figure 3f. The reflection spectrum of the polished surface is provided in black as a reference. It is apparent that the reflection of the sample after laser treatment was greatly restrained. The reflectance of the sample with an interval of 90 μm was lower than 10% over the wavelength ranging from 250 nm to 2250 nm. When the scanning interval was further decreased, the antireflection of the sample could be improved to be less than 7% (at an interval of 50 μm). The deep air holes distributed on the metal surface are considered to have served as light-trapping spots that were the dominant factor enhancing the antireflection and limiting the scattering. When the scanning intervals were decreased to be 30 μm, the reflectance appeared to slightly increase. We attribute this phenomenon to the decrease in the number and depth of the microholes, which could weaken the multireflection processes.

### 3.2. Effect of Nanosecond Laser Scanning Velocity on Surface Reflectance

The morphologies of the shaped microstructures fabricated on TC4 surfaces using different nanosecond laser scanning velocities with a definite interval of 50 μm are illustrated in Figure 4. Figure 4a–c show samples with scanning velocities of welding lines ranging from 6 mm/s to 100 mm/s, and Figure 4d–f shows local magnifications of Figure 4a–c, respectively. Compared with the surface with a scanning velocity of 25 mm/s, fewer microholes were observed in the laser-irradiated region in Figure 3b,e. When the scanning velocity was increased to 50 mm/s, microgrooves clearly appeared, and micron stripes gradually emerged on the bottom of the microgrooves. However, regular microgrooves provided fewer multireflection processes and had a weak ability to capture light scattering. In contrast, when the scanning velocity was decreased to 6 mm/s, the massive accumulation of energy raised the temperature within the laser-irradiated region, triggering more intense ablation. The jets induced by the ablation covered the sample surface and smoothed the concavo-convex microstructures, inhibiting the establishment of microholes.

Figure 5 shows the reflection spectra of the samples with different laser scanning velocities ranging from 6 mm/s to 100 mm/s. The reflection spectrum of the polished surface is provided in black as a reference. It can be observed that the reflection levels of the samples with different laser scanning velocities exhibited different degrees of reduction. The reflectance of the sample with a scanning velocity of 100 mm/s was merely lower than 32% over the wavelength ranging from 250 nm to 2250 nm. When the scanning velocity was decreased to 50 mm/s, the reflection of the sample was further inhibited to be less than 17%. The antireflection of the sample reached its best result at the scanning velocity of 25 mm/s, which is provided as a blue curve, and the blue curves in Figure 5 and Figure 3f represent the reflectance of the same sample. When the scanning velocity was further decreased to 6 mm/s, the reflectance showed a certain increase. The cause of this phenomenon is that the concavo-convex surfaces covered with re-solidified metal did not provide enough light-trapping structures to suppress reflection.

### 3.3. Effect of Nanosecond Laser Pulse Energy on Surface Reflectance

The maximum energy of the experimental laser was employed in the above study, and we used the other three laser energy parameters to fabricate TC4 samples to explore the pulse energy effects on the surface reflectance. The reflection spectra of the samples with pulse energies ranging from 200 μJ to 50 μJ are shown in Figure 6. As the laser pulse energy gradually decreased to 50 μJ, the reflectance of the sample eventually increased to 14% over the wavelength range from 250 nm to 2250 nm. Under different pulse energies, the trend of reflectance with wavelength was basically the same: rapidly rising to a certain intensity in the ultraviolet band, staying reasonably steady at a certain intensity in the visible spectrum, and slowly climbing again in the near-infrared band. Within the range of our experimental conditions, higher laser pulse energies produced a lower spectral reflection.

### 3.4. Effect of Femtosecond Laser Pulse Modification on Surface Reflectance

The parameters of nanosecond laser fabrication were studied in detail above, and this allowed us to select the best antireflection sample as a foundation for the femtosecond laser processing. The focused femtosecond laser with extreme intensities in the TW/cm^2^ range is generally recognized to be good at inducing nanostructures during laser surface ablation. Considering the relatively short pulse duration (<10^−12^ s), femtosecond laser processing is applied to build micro-nanostructures due to its characteristic of quasi-nonthermal processing. To explain the formation of the femtosecond-laser-treated stripes, the surface plasmon polaritons (SPPs) has been proposed as the mechanism responsible for the surface wave generation in metals. The interference between the electromagnetic field of the SPPs and the incident laser pulse leads to a spatially modulated deposition of optical energy to the electronic system of the material [41,42,43,44,45].

The sample pretreated by nanosecond laser (fixed velocity at 25 mm/s and interval of 50 μm) was further irradiated using the focused femtosecond laser with orthogonal-direction scanning. The reflection spectra of post-treated samples with femtosecond laser scanning velocities ranging from 6 mm/s to 100 mm/s are shown in Figure 7a. The reflection spectrum of the nanosecond-laser-treated base is provided in black as a reference. It is apparent that the reflection of the sample after femtosecond laser treatment was obviously suppressed over most of the measurement wavelength range from 250 nm to 2250 nm. When the femtosecond laser scanning velocity was set to 8 mm/s, the reflection spectrum of the sample had a very significant peak in the range of 250–700 nm. As the scanning velocity was moderately increased, the peak value in the spectra was gradually weakened. The reflectance of the sample with a scanning velocity of 12 mm/s attained the minimum value, with an average reflectance of 3.1%. When the scanning velocity was further increased to 14 mm/s, the reflectance increased. 

The morphology of the post-treated sample with a femtosecond laser scanning velocity of 12 mm/s was stepwise magnified, as illustrated in Figure 7b–e. The modified sample surface was scratched into grids of 50 × 50 μm, as shown in Figure 7b. Figure 7c–d shows that the femtosecond laser treatment induced deeper air holes and nanostripes on the microstructures. Deeper holes enhance the internal multi-reflection process. Nanostripes are observed to be distributed perpendicular to the scanning direction with an average width of 330 nm. In the air hole structures, the nanostripes distributed on the walls of the holes can effectively collect the incident light in a broad spectrum and from all directions. These two aspects jointly promote the light-trapping effect of air holes. Meanwhile, it is well known that nanoscale structures can generate intense surface plasmon resonance (SPR) absorption [46,47,48]. The aggregation of the nanostripes can lead to a broadening effect of the resonance bands to form broadband antireflection. By superimposing the light-trapping effect of air holes and the optical impedance matching effect of nanostripes, ultralow reflectance of the surface of the hybrid structures was realized.

## 4. Conclusions

In our study, we proposed and experimentally demonstrated a two-step strategy for structuring the surface of TC4 using femtosecond and nanosecond lasers and successfully realized ultralow broadband interface reflection. The morphological characterization proved that deep air holes were induced on the sample surface at the microscale, and these played a predominant role in the antireflection. TC4 with this kind of native antireflection is pollution free and useful to eliminate scattering when it serves as an optical tube or holder. The femtosecond-laser-induced nanostructures on the walls of air holes further enhanced the multi-reflection processes, which obtained a satisfying antireflection result. The reflection of the sample covered by dual-scale structures was determined to be less than 5% over the wavelength range from 250 nm to 2250 nm. This two-step strategy is believed to be a highly efficient option for constructing functional surfaces.

## Figures and Tables

**Figure 1 micromachines-11-00020-f001:**
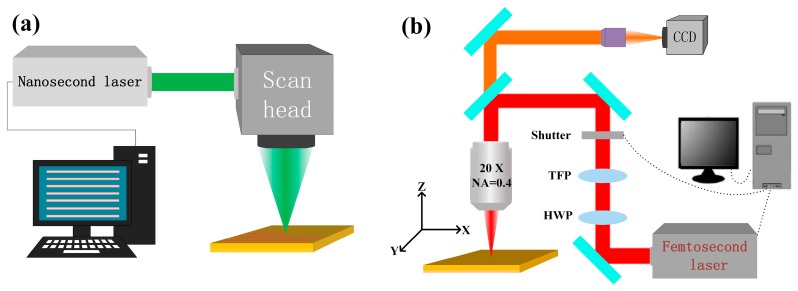
Schematic of the two-step nanosecond and femtosecond laser setup for fabrication of micro-nano hybrid structures on Ti-6Al-4V (TC4) surfaces: (**a**) nanosecond laser fabrication system and (**b**) femtosecond laser fabrication system.

**Figure 2 micromachines-11-00020-f002:**
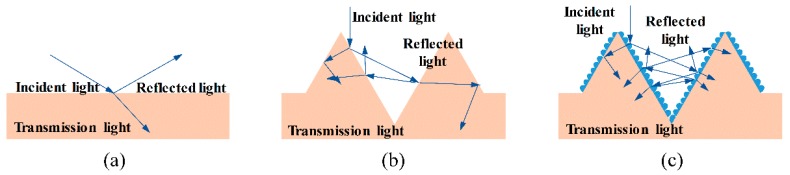
A schematic diagram of light wave propagation on sample surfaces with different morphologies: (**a**) polished surface, (**b**) surface with microstructures, and (**c**) surface with hierarchical micro- and nanostructures.

**Figure 3 micromachines-11-00020-f003:**
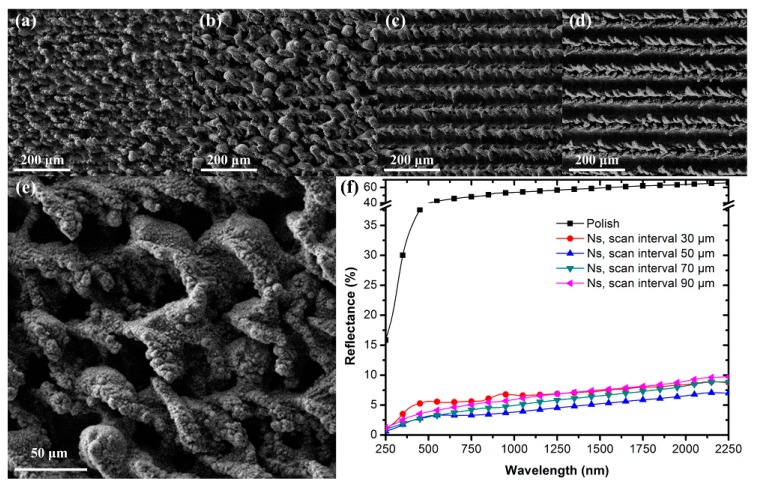
SEM images of the microstructures characterizing different scan intervals of (**a**) 30 μm, (**b**) 50 μm, (**c**) 70 μm, and (**d**) 90 μm, respectively. (**e**) High-magnification SEM image corresponding to (b). (**f**) Comparison of the surface reflectance values corresponding to different intervals.

**Figure 4 micromachines-11-00020-f004:**
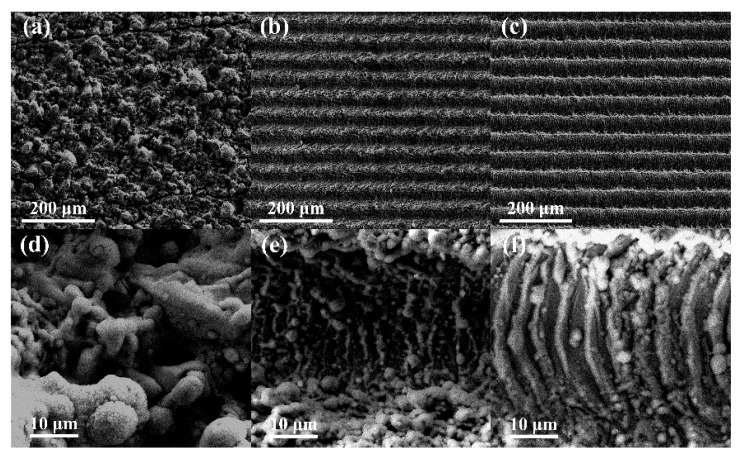
SEM images of the microstructures characterizing different scanning velocities of (**a**,**d**) 6 mm/s, (**b**,**e**) 50 mm/s, and (**c**,**f**) 100 mm/s; (**d**)–(**f**) are high-magnification SEM images corresponding to (**a**)–(**c**), respectively.

**Figure 5 micromachines-11-00020-f005:**
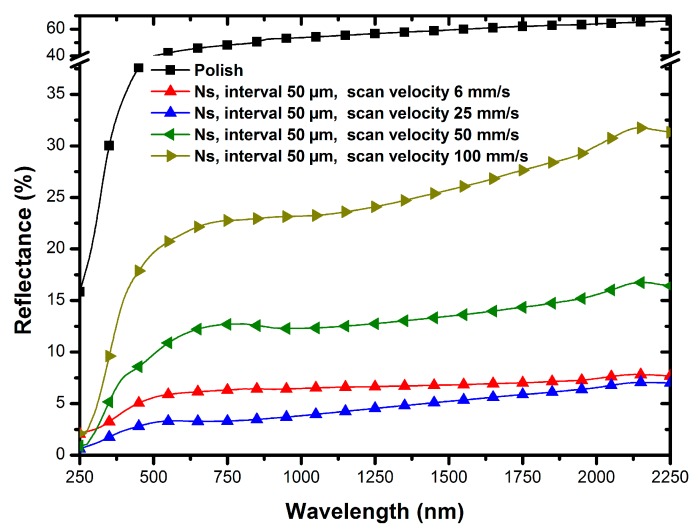
Comparison of the surface reflectance of samples from different scanning velocities.

**Figure 6 micromachines-11-00020-f006:**
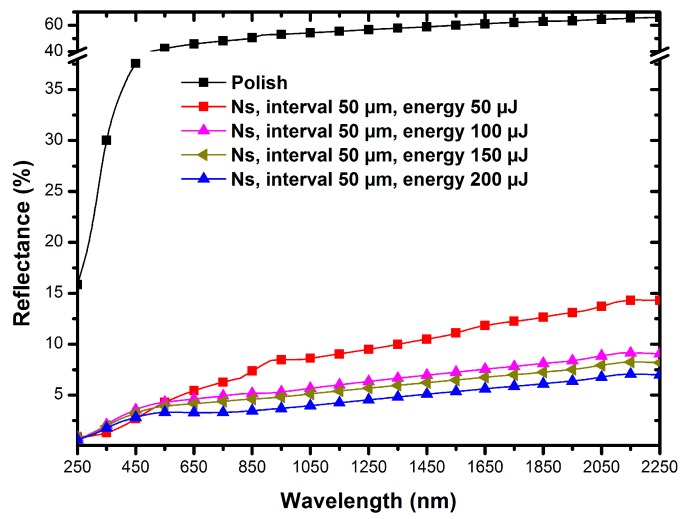
Evolution of the surface reflectance of the microstructures fabricated using different pulse energy levels.

**Figure 7 micromachines-11-00020-f007:**
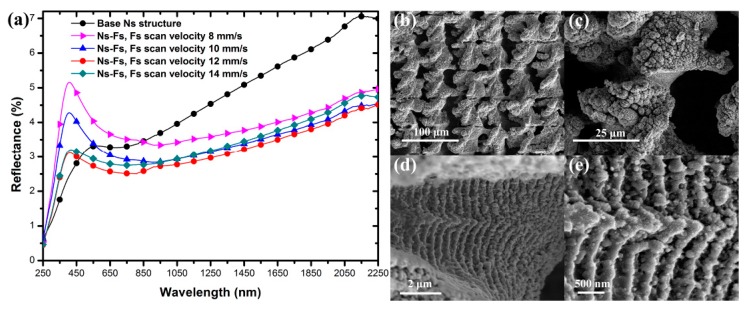
(**a**) Evolution of the surface reflectance of the micro-nano hybrid structures modified using different femtosecond laser scanning velocities. (**b**–**e**) SEM images of the stepwise-magnified micro-nano hybrid structures with optimum femtosecond laser modification parameters.

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
