# Peer review of "Design and Fabrication of Dual-Scale Broadband Antireflective Structures on Metal Surfaces by Using Nanosecond and Femtosecond Lasers"

_micromachines, 2019, doi:10.3390/mi11010020_

Round 1

Reviewer 1 Report

Major:

1. The literature review is rather limited. The recent advances and pioneering works of the antireflective surface laser texturing are ignored [DOI: 10.1038/244281a0; DOI: 10.1002/adma.201901123; DOI: 10.2961/jlmn.2015.02.0004]. I recommend extend literature review by including some in depth papers dedicated to absorptivity enhancement and antireflective surfaces created by laser texturing.

2. The reflection and absorption is mentioned in the manuscript. However, there is more physical phenomenon like absorption, transmission, scattering, and reflection as equation A+T+S+R=1 is always valid. I understand that transmission for thick metal is zero (T=0), however, scattering might be large for micro/nanostructures on metal surface. The “scattering” is mentioned only once in the conclusions of the manuscript. I recommend include more discussion related to the scattering in to the text of the manuscript.

3. Laser fabrication and micro/nano structuring rate is deepened on laser fluence applied to the TC4 material; therefore, characterization of laser energy density on the sample in J/cm^2 is highly important. For fluence characterization the exact spot sizes on sample are needed. The spot size of nano-second laser beam of 10 um is provided; however, for femtosecond beam spot size is not given. Moreover, no measurements technique is described in paper. There is special technique called D2 versus Fluence method [DOI: 10.1364/OL.7.000196; DOI: 10.1038/s41598-018-35604-z] for direct measurement of laser spot size on the sample. Or similar technique of ablated area versus laser fluence technique can be applied [DOI: 10.1016/j.solener.2014.01.013; DOI: 10.1364/CLEO_AT.2018.AM1M.2]. Please, provide data how beam size was characterized in this work and what are measurement errors.

4. The laser ablation threshold is the most important characteristics for characterization of the laser ablation process. Ablation threshold is not provided for TC4 material in the manuscript. Please provide ablation thresholds for nanosecond and femtosecond beams ant compare to literature values if available.

5. The laser texturing rate or ablation rate has a maximum value, predicted theoretically with peak laser fluence F0 = e2 X Fth ≈ 7.4 X Fth, where Fth [J/cm2] is ablation threshold [DOI: 10.2961/jlmn.2009.03.0008; DOI: 10.1063/1.2794376]. This fact has been recently confirmed experimentally in scientific papers [DOI: 10.1117/12.846521; DOI: 10.1016/j.optlaseng.2018.11.001]. However, in presented paper laser fluence optimization presented is not presented. In fact, in this paper only two-step hierarchical texturing experimental data is given. However, no texturing speeds in mm^2/s or similar processing rate is not given (only beam scanning velocities of 6-100 mm/s for nanosecond laser and 8 mm/s for femtosecond laser are given). It might be interesting to see the structuring rates of two-step process (nanosecond and femtosecond). The evaluation of surface texturing price (Dollar per square meter) of this type of processing would be also interesting for commercialization of proposed technology. I also recommend give spot size or pulse energy optimization in order to achieve maximal available texturing rate.

6. The reflectance dependent on the wavelength is given in the fig3, fig5, fig6, and fig7. The laser self-organized structures have great antireflective properties over wide angle of incidence [DOI: 10.1002/adma.201901123]. I recommend include reflectance measurement depending on the incidence angle at fixed wavelength, and see how those structures behave at different angles, if such equipment is available in the laboratory.

7. In fig3, fig5, fig6, and fig7 the undefined processing codes are given for different processing regimes as “NS laser stru. P30”, or “NS laser stru. P50 V6”, or “Hybit Stru. V10” etc. Those CODES give no information about physical parameters, that have been used for each of the test. I recommend including physical parameters with units and full description in to the graphs l, like “NS, scan speed 100 mm/s” or similar.

8. Authors claim that surface morphology and air pockets are responsible for the antireflective properties. However, chemical composition of laser structured areas is not investigated. Maybe surface chemistry changes during laser processing and strong oxidation occurs during nanosecond laser structuring in air [DOI: 10.1016/j.promfg.2019.06.149]. I recommend including discussion on this point.

9. The “Fresnel-equation-based mathematical model” (line 115) is mentioned. However, no model equations are given. Please explain.

10. Please give more detail explanation on figure 2b and figure 2c, why hierarchical micro nano structures are more effective than microgrooves.

Minor:
11. In the reference [2] the year of publication is not correct “200”, and has to be replaced by correct year “2000”.

12. In the reference [18] page numbers are missing.

13. In the reference [24] the space singe is missing “Optics Express2011” between journal name and year.

14. The reference number “[A.]” (line 316)has to be replaced by number [25].

15. In the [A.] reference the journal name “JOURNAL OF PHYSICAL CHEMISTRY B“ (line 317) has to be replaced by small letters “Journal of Physical Chemistry B“.

16. The sample size “25×25×10 mm” in the materials description has to be replaced by the “25 mm ×25 mm ×10 mm” or by the “25×25×10 mm^3”.

17. In the reference [24] the space singe is missing “Nature Communications2011” between journal name and year.

Reviewer 2 Report

General remarks:

In order to be suitable for publication, the authors must perform an intense major revision of the language of the manuscript. Otherwise the manuscript should not be considered. There are quite many errors, especially regarding the verb tenses, but not solely.

Starting with the title, which redaction should be revised: “…nanosecond and femtosecond lasers.”

Abstract: Lines 26-28: “The antireflection mechanism of the dual-scale structures is discussed by utilizing kinds of characterizations”. Please rephrase the sentence.

Introduction: line 36: “The hierarchical micro and nanostructures, inspiring by nature,…” should be “The hierarchical micro and nanostructures, inspired by nature,

Or line 41: “For instance, the ZnO substrates were covered by biomimetic hierarchical structures exhibits low reflection over a wide spectral range” should say “For instance, ZnO substrates covered by biomimetic hierarchical structures exhibits low reflection over a wide spectral range.

Line 264: Erratum in the reference number 2 (missing year). There are also errata in lines 314, 316, 317.

And a large etcetera.

Comments on the INTRODUCTION section:

* The authors should add more recent references related to the fabrication of anti reflective surfaces for the different applications cited in the introduction. It seems that those applications are no longer interesting within the scientific community. Now, anti reflective surfaces are more focussed on optical applications in the fields of solar energy components, optical elements, and optoelectronic devices.

* Recently, and among other works, Fan et al. reported the fabrication of antireflective surfaces on Ti with average reflectance values below 2,4% using just the femtosecond laser regime (doi.org/10.1021/acsnano.7b03673).

Therefore, a clear discussion with references on why the combination of nanosecond and femtosecond regimes contributes to the improvement of the antireflective surfaces should be addressed to validate the novelty of both the results and the method.

Reviewer 3 Report

Unfortunately, this paper should be seriously improved from the scientific point of view, and on the writing form and the English grammatical/style. For this reason, I recommend a major revision.

Please see the attached file for my comments.

Round 2

Reviewer 1 Report

Dear authors,

Thank your for your effort in revising the manuscript according to the comments. My recommendation is accept after minor text editing revision.

Minor revision:

I recommend replace incorrect phrase "with using" by correct phrase "by using" in the title of manuscript "Design and fabrication of dual-scale broadband antireflective structures on metal surfaces BY using nanosecond and femtosecond laser".

Author Response

Point 1: I recommend replace incorrect phrase "with using" by correct phrase "by using" in the title of manuscript "Design and fabrication of dual-scale broadband antireflective structures on metal surfaces BY using nanosecond and femtosecond laser".

Response 1: We thank the reviewer for the valuable comments on our manuscript. We had replaced incorrect phrase "with using" by correct phrase "by using" in the title of manuscript "Design and fabrication of dual-scale broadband antireflective structures on metal surfaces BY using nanosecond and femtosecond laser".

Reviewer 3 Report

The paper was completed on some sections in the resubmited paper but the fundamental part on laser interaction as I proposed in my previous review have not been considered (works from Bulgakova, T. Itina....).

The paper remains more at a description of a set of experiments and results more thant a sounding scientific paper.

This crucial fundamental point (and English style) should be completed rewritten.

Round 3

Reviewer 3 Report

The paper was significantly improoved. Remains few syntax/style should be corrected